# Soil Reservoir Dynamics of *Ophidiomyces ophidiicola*, the Causative Agent of Snake Fungal Disease

**DOI:** 10.3390/jof7060461

**Published:** 2021-06-08

**Authors:** Lewis J. Campbell, Joanna Burger, Robert T. Zappalorti, John F. Bunnell, Megan E. Winzeler, Daniel R. Taylor, Jeffrey M. Lorch

**Affiliations:** 1United States Geological Survey—National Wildlife Health Center, Madison, WI 53711, USA; mwinzeler@usgs.gov (M.E.W.); Daniel.Taylor3@usda.gov (D.R.T.); 2Department of Pathobiological Sciences, School of Veterinary Medicine, University of Wisconsin-Madison, Madison, WI 53706, USA; 3Division of Life Sciences, Environmental and Occupational Health Sciences Institute, and Pinelands Research Station, Rutgers University, Piscataway, NJ 08854, USA; burger@dls.rutgers.edu; 4Herpetological Associates, Inc., Pemberton, NJ 08068, USA; rzappalort@aol.com; 5New Jersey Pinelands Commission, New Lisbon, NJ 08068, USA; john.bunnell@pinelands.nj.gov

**Keywords:** disease mitigation, environmental persistence, environmental reservoirs, fungal pathogens, pathogen inhibition, emerging infectious disease

## Abstract

Wildlife diseases pose an ever-growing threat to global biodiversity. Understanding how wildlife pathogens are distributed in the environment and the ability of pathogens to form environmental reservoirs is critical to understanding and predicting disease dynamics within host populations. Snake fungal disease (SFD) is an emerging conservation threat to North American snake populations. The causative agent, *Ophidiomyces ophidiicola* (Oo), is detectable in environmentally derived soils. However, little is known about the distribution of Oo in the environment and the persistence and growth of Oo in soils. Here, we use quantitative PCR to detect Oo in soil samples collected from five snake dens. We compare the detection rates between soils collected from within underground snake hibernacula and associated, adjacent topsoil samples. Additionally, we used microcosm growth assays to assess the growth of Oo in soils and investigate whether the detection and growth of Oo are related to abiotic parameters and microbial communities of soil samples. We found that Oo is significantly more likely to be detected in hibernaculum soils compared to topsoils. We also found that Oo was capable of growth in sterile soil, but no growth occurred in soils with an active microbial community. A number of fungal genera were more abundant in soils that did not permit growth of Oo, versus those that did. Our results suggest that soils may display a high degree of both general and specific suppression of Oo in the environment. Harnessing environmental suppression presents opportunities to mitigate the impacts of SFD in wild snake populations.

## 1. Introduction

Emerging infectious diseases (EIDs) are being documented at a faster rate during the 21st century than at any prior point in recorded history [1], representing an ever-increasing global threat to human [2], domestic [3] and wild animal [4], and plant health [5]. Within host populations, the dynamics of EIDs are governed by interactions between hosts, the causative pathogen, and the environment [6,7]. Particularly, the ability of a pathogen to exist outside of its host can have critical implications for disease dynamics [7,8]. This is because the fitness of pathogenic organisms is fundamentally a trade-off, balancing within-host replication and between-host transmission [9]. Highly virulent pathogens that have a strategy focused on within-host replication may have a lower potential for between-host transmission because afflicted hosts often have reduced mobility and contacts with other susceptible hosts [6]. However, a pathogen that is able to exist outside of its hosts, in environmental reservoirs or secondary host or vector species, is liberated from this balancing selection [10]. The “sit-and-wait” hypothesis postulates that the existence of a stable environmental pathogen reservoir allows a disease to escape the constraints of density-dependent transmission and attain high levels of virulence [11,12,13,14]. Because of this, pathogens that can effectively form reservoirs pose a greater risk of extinction to their host populations [4]. Understanding the potential of a given pathogen to form environmental reservoirs is, therefore, critical in determining the threat that a pathogen poses to host populations and devising effective mitigation strategies.

Snake fungal disease (SFD) or ophidiomycosis is a fungal EID that impacts numerous species of North American snakes [15]. It is characterized by skin lesions, which, in severe cases, can result in death [15]. The first confirmed cases of SFD occurred in a population of eastern massasaugas (*Sistrurus catenatus catenanus*) in Illinois during 2008 [16]. The disease has since been observed in a number of species throughout the eastern half of the USA [15,17]. The drivers of SFD emergence are not well understood [15]. However, it is considered a significant threat to many snake populations [15,16,18]. Snake fungal disease is most often observed in animals immediately following their emergence from subterranean hibernation [15]. It is not currently known whether this is due to an increased risk of infection during hibernation or due to the progression of already established infections during the hibernation period [19].

The fungal pathogen *Ophidiomyces ophidiicola* (Oo) is the causative agent of SFD [20]. *Ophidiomyces*
*ophidiicola* is detectable in environmental samples [21] and is capable of growing on a wide variety of substrates in vitro [19]. These observations have led to speculation that Oo is capable of establishing environmental reservoirs [19]. However, little is known about the dynamics of the fungus in natural substrates, such as soils. Soils have long been known to harbor environmental reservoirs of pathogenic microbes, especially those which impact plants [22] and humans [23]. Many fungal and bacterial pathogens are capable of replication in soil as saprotrophs, while others, particularly fungi, may form dormant, environmentally persistent propagules [24]. Growth and persistence of these microorganisms are linked to various parameters of the soil such as pH, nutrient content, and soil structure [25,26]. Research focused on pathogens of agricultural crops has shown that biotic factors of soils, such as other microbial species that exist in soil communities, can affect the ability of those soils to act as pathogen reservoirs [5,27]. Competition for resources or directly antagonistic interactions with other microbial species can reduce the survival of pathogenic organisms [27]. Conversely, the same interactions can prevent the germination of dormant life stages, increasing the duration of environmental persistence [24]. Due to the conservation threat that Oo poses to North American snake species [18], a greater understanding of where the fungus is found in the environment, its ability to form environmental reservoirs, and the importance of environmental reservoirs in SFD ecology is critical.

Here, we investigated the potential for Oo to establish environmental reservoirs in areas frequented by its snake hosts. Specifically, we examined the prevalence of Oo in sediment samples collected from within and above underground snake dens in an experimental field system in New Jersey (USA), conducted soil microcosm growth assays to better understand the ability of Oo to grow and persist in the environment, and used next-generation sequencing techniques to classify the bacterial and fungal microbial communities that may influence the ability of soil to serve as a reservoir for Oo. We hypothesized that: (A) the prevalence of Oo would be higher in soils sampled from within snake hibernacula than in topsoil; (B) Oo would proliferate in soils, but that proliferation would be greater in soils samples collected from within hibernacula; and (C) increased Oo growth would be linked to soil characteristics such as pH and the composition of native microbial communities.

## 2. Materials and Methods

### 2.1. Initial Survey

In February and March 2018, as part of ongoing SFD surveillance, 15 soil samples were collected from within 3 snake hibernacula (dens 1–3) in New Jersey (Burlington and Ocean Counties) that contained primarily northern pine snakes (*Pituophis melanoleucus melanoleucus*). Northern pine snakes are a threatened species in New Jersey and more specific localities are not provided due to concerns over illegal harvest. The hibernacula chambers of these dens are artificially excavated annually as part of ongoing, long-term ecological studies of northern pine snakes [28,29,30,31] and are accessible while snakes are hibernating. Besides annual excavation, hibernacula are not modified in any way. A soil sample was collected from immediately underneath each snake found to be hibernating within each excavated hibernaculum.

DNA was extracted from 300 mg of each soil sample with the Qiagen (Hilden, Germany) DNeasy PowerLyzer PowerSoil Kit using a modified protocol. Specifically, soils were added to 2 mL microcentrifuge tubes containing 0.1 mm glass beads and a solution of 750 µL of the kit-provided powerbead solution, 60 µL of kit-provided buffer C1 and 20 µL of 50 U/µL lyticase solution. Lyticase solution was created by dissolving lyophilized lyticase (Sigma Aldrich, St. Louis, MO, USA) in extraction kit-provided powerbead solution. Soils were incubated at 30 °C for 30 min, before bead beating at 4 kHz for 45 s. Extraction then proceeded following manufacturers’ protocols and DNA was eluted into 100 µL of EDTA buffer (Invitrogen, Carlsbad, CA, USA). Extractions were screened using a real-time PCR assay specific for the internal transcribed spacer region of Oo [32]. Standard curves were constructed using synthetic gBlocks oligos matching the target region (IDT, Coralville, IA, USA). These standard curves consisted of six 10-fold serial dilutions ranging from 35 to 35,000,000 copies. Each standard dilution was run in triplicate on each qPCR plate. Any soil sample that returned a cycle threshold (CT) value less than or equal to 40 (maximum cycle number) was considered positive.

A subset of four soil samples that tested positive for Oo by real-time PCR was screened for the presence of viable Oo using a dilution plating method as described in [33]. Briefly, ~200 mg of soil was suspended in 500 µL sterile water and vortexed. Suspended soils were serially diluted ten-fold down to a 1000-fold dilution. One hundred fifty microliters of the undiluted suspension and each subsequent ten-fold dilution were plated onto dermatophyte test medium (DTM) in duplicate. The plates were incubated at 30 °C for 15 days, at which time colonies exhibiting a morphology consistent with Oo were isolated onto fresh DTM. Identification of isolated colonies was confirmed by sequencing the internal transcribed spacer region as described previously [20].

As Oo was detected in these soil samples by both PCR and culture (see Section 3), additional sampling and experiments were conducted as outlined below. Data from the initial survey were not included in any further analysis.

### 2.2. Sample Collection

In March 2019, soils were collected from one of the snake hibernacula sampled during the initial survey and an additional four hibernacula from the same New Jersey Pinelands. One ~1.5 kg soil sample was collected from each excavated hibernation chamber. Each internal soil sample, henceforth referred to as hibernaculum samples, was matched with a topsoil sample (topsoil) taken from within 1 m of the den entrance. As all sampled hibernacula were located in the same pine barrens habitat, the above ground landscape surrounding each hibernaculum was similar. For the remainder of the manuscript, the term “den” refers to the collective location of paired hibernaculum and topsoil samples (i.e., den 1 hibernaculum and den 1 topsoil). A total of six hibernaculum and four topsoil samples were collected (den 4 was represented by two hibernaculum soils due to the presence of two distinct hibernation chambers and a single topsoil sample represented both den 5 and 6 as these two independent dens were located in close proximity). Soil samples were collected by scooping soil with sterile wooden spatulas into sterile 500 mL plastic jars with screw-cap lids (Nalgene, Rochester, NY, USA). Soils were stored chilled (4 °C) until processing. Hibernating snakes were present in all excavated hibernacula.

### 2.3. Environmental Oo Detection

To assess the prevalence of Oo in soil samples, DNA was extracted from three 300 mg aliquots of each soil as described above. The amount of Oo DNA present in each soil aliquot was determined using quantitative PCR as described above. We compared the number of detections between topsoil samples and internal hibernaculum soils using a Fisher’s exact test of contingency tables, implemented in R [34].

### 2.4. Soil Sterilization and Characterization

A 1000 g sample of each of the 10 experimental soils was divided into two 500 g portions in glass beakers. One 500 g beaker of each soil was sterilized by autoclaving. Soils were autoclaved twice for 90 min, at 15 psi and 121 °C, with a 24 h gap between each autoclaving cycle. A 300 mg aliquot of each soil was suspended in 1 mL filter-sterilized (0.2 µm) phosphate buffered saline with added 0.5% Tween20 (PBST); 100 µL of this suspension was plated directly onto a Sabouraud dextrose agar plate. Plates were incubated at 24 °C in the dark for 7 d. Sterility of each soil was confirmed if no microorganisms grew on plates.

To characterize soil pH, phosphorus, potassium and organic matter content, approximately 350 g of the 20 soil samples (10 autoclave-sterilized and 10 non-sterile) was sent to the University of Wisconsin-Madison’s soil sciences extension facility (Marshfield, WI, USA). Following testing for the assumptions of parametric statistical tests, the impact of autoclaving on soil parameters was compared using paired *t*-tests between soil samples before and after autoclaving. Soil parameters were compared between samples using one-way *t*-tests. All statistical analyses were conducted using R v3.5.1 [34].

### 2.5. Microcosm Growth Assays

We investigated the persistence and proliferation of Oo in environmental soil samples using in vitro microcosm growth assays. Assays were performed in both sterile and non-sterile soil from all ten samples collected in 2019. Soils were aliquoted into 300 mg microcosms in 2 mL screw-cap microcentrifuge tubes (Thermo Fisher Scientific, Waltham, MA, USA). We created replicate microcosms for each experimental soil and treatment combination (as outlined below), allowing for three time points and three replicates of each combination at each time point. For each soil, the following combinations were created at each of the three time points: sterile soil spiked with live conidia (*n* = 3), sterile soil spiked with heat-killed conidia (*n* = 3), sterile soil spiked with PBST only (*n* = 3), non-sterile soil spiked with live conidia (*n* = 3), non-sterile soil spiked with heat-killed conidia (*n* = 3), and non-sterile soil spiked with PBST only (*n* = 3).

Live Oo conidia (from isolate NWHC 24564-1) were harvested from a 14-day-old culture following the protocol of Lorch et al. [20,35]. Conidia were suspended in filter-sterilized PBST, enumerated using a hemocytometer, and diluted with additional PBST to a concentration of 25 conidia per microliter. The suspension was then divided: half was immediately used as inoculum (live conidia) to spike soil microcosms, while the other half of the conidial solution was inactivated (heat-killed conidia) by emersion into a boiling water bath for 30 min. We inoculated soil microcosms with 40 µL of conidial suspension containing 1000 either live or heat-killed conidia. A batch of microcosms serving as a control group was established by spiking additional replicates of each soil with filter-sterilized PBST containing no fungal conidia. Immediately post-inoculation, the first three replicates of each soil and treatment combination were stored at −80 °C to serve as time point 1 (0 days post-inoculation). Remaining microcosms were incubated at 24 °C in the dark. Three more replicates of each combination were placed at −80 °C (to stop growth) at 15 days post-inoculation; the same was performed for the final three replicates of each combination at day 30 post-inoculation. DNA was extracted from experimental microcosms following the same protocol outlined above. The amount of Oo in each microcosm was again quantified using the qPCR assay [32]. Due to small within replicate group sample sizes (*n* = 6) the growth of Oo was qualitatively determined by upward trajectory of growth curves and quantitively determined by a 2 order of magnitude (or greater) difference in mean Oo copy number (determined by comparison with standard curve) between live-spiked and heat-killed treatments for a given soil in both sterile and unsterile state at each time point. No control microcosms for any soil which was not already known to contain naturally occurring Oo yielded a positive qPCR result, and the abundance of Oo in these microcosms did not increase during the experiment; thus, control group data were excluded from further analyses.

Colony counts were performed as a secondary method to assess growth and viability of Oo in the microcosms. To achieve this, an additional three microcosm replicates inoculated with viable conidia were created for each soil at 0 d and 30 d post-inoculation. Prior to plating, these microcosms were suspended in 1000 µL of PBST and serially diluted from 10^−1^ to 10^−3^. For each resulting dilution, 100 µL was plated onto DTM agar plates. Plates were incubated for 15 d at 24 °C in the dark at which point Oo colony forming units (CFU) were enumerated.

### 2.6. Microbial Community Analyses

We characterized the bacterial and fungal communities present in our 10 experimental soils using amplicon sequencing. The 16S rRNA (v4) and internal transcribed spacer (ITS2) amplicons were sequenced for each sample following the protocols described in Appendix A. Following sequencing, we investigated how patterns in soil microbial community richness, diversity and composition correspond with the soil type (hibernaculum vs. topsoil), environmental Oo detection, and experimental Oo growth, following the methods outlined in Appendix A. Briefly, microbial community analyses were primarily conducted using the R package phyloseq [36]. We computed diversity and richness statistics for the microbial communities of all soils as well as comparing the microbial community composition of each soil using non-metric multidimensional scaling. We investigated how microbial community richness and diversity related to the detection and growth of Oo in soils using generalized linear mixed models. We used permutational analysis of variance to compare microbial community composition of soils that were naturally Oo positive and those that were not, and those in which Oo grew during experimental microcosm experiments and those in which it did not.

## 3. Results

### 3.1. Initial Survey

Of the 15 hibernaculum soil samples collected and tested for Oo presence by qPCR in 2018, 13 (86%) were found to be positive (Table 1). Two of the four soil samples screened for the presence of viable Oo using culture techniques yielded Oo isolates (Table 1). Identification of the isolates was confirmed based on 100% sequence identity of the ITS region with other isolates of Oo present in GenBank.

### 3.2. Oo Is More Prevalent Within Snake Hibernacula

Quantitative PCR detected Oo in 11 out of 18 (61%) hibernaculum soil samples collected in 2019 and only one of 12 (8%) topsoil samples (Table 1). A Fisher’s exact test of contingency tables showed this to be statistically significant (*p* = 0.007, odds ratio = 0.064).

### 3.3. Oo Is Capable of Growth in Sterile Soils

Quantitative PCR demonstrated that growth occurred in microcosms of four out of 10 soils that had been sterilized prior to inoculation (Figure 1). Of these four soils, three were topsoil samples and one was a hibernaculum soil (Figure 1). The results of CFU plate counts supported qPCR data with CFUs increasing between day 0 and day 30 for three of the four soils in which qPCR growth was observed (Appendix A). No growth was observed in any soil in which the native microbial community was still active (Figure 1).

### 3.4. No Evidence That Detection or Growth of Oo Is Linked to Abiotic Soil Parameters

The results of parameter analysis of our experimental soils are presented in Appendix A. Statistical analysis showed that hibernaculum soils have significantly lower potassium levels (T = −3.79, DF = 3.33, *p* = 0.03) and organic matter content than associated topsoil samples (T = −3.04, DF = 3.35, *p* = 0.049). Paired *t*-tests showed that autoclaving significantly lowered the pH of experimental soils from a mean of 5.15 (±0.6) to 4.12 (±0.37, T = 4.6161, DF = 14.887, *p* ≤ 0.01). No other parameter was impacted by autoclaving. The environmental detection and experimental growth of Oo in experimental soils was not correlated with any measured parameter of the soils.

### 3.5. Detection and Growth of Oo Are Correlated with Microbial Community Diversity but Not Richness

The results of our microbial community classification via amplicon sequencing are presented in full in our Appendix A. Here, we present the results of comparisons between our experimental soils.

Bacterial communities of hibernaculum soils were significantly less diverse than topsoil samples based on effective number of species scores (ENS; Est = 3.2, CI = 1.48–4.91, *p* ≤ 0.001, Figure 2A). Bacterial species richness was also significantly higher in topsoil samples (observed genera, IRR = 3.38, CI = 1.18–9.63, *p* = 0.02, Figure 2B). Fungal communities of hibernaculum soils were also significantly less diverse (ENS; Est = 0.84, CI = 0.34–1.34, *p* = 0.001, Figure 2C) and species rich (observed genera, IRR = 17.50, CI = 3.82–80.16, *p* ≤ 0.001, Figure 2D) than topsoil samples.

The environmental detection of Oo was associated with significantly higher alpha diversity of fungi (ENS; Est = 0.75, CI = 0.14–1.35, *p* = 0.016, Figure 2C) and fungal species richness (observed genera, IRR = 7.44, CI = 1.40–39.43, *p* ≤ 0.018, Figure 2D). Environmental detection of Oo was not correlated with bacterial species diversity (ENS; Est = 1.09, CI = −0.69–2.87, *p* = 0.23, Figure 2A) but was associated with increased richness (observed genera, IRR = 9.02, CI = 2.67–30.39, *p* ≤ 0.001, Figure 2B).

Soils in which Oo grew during microcosm experiments had significantly higher bacterial diversity based on ENS prior to sterilization; (ENS; Est = 2.58, CI = 1.42–3.74, *p* ≤ 0.001, Figure 2A) and bacterial species richness (observed genera; IRR = 6.84, CI = 2.33–20.09, *p* ≤0.001, Figure 2B). The growth of Oo in experimental soils was not correlated with diversity of fungal species prior to sterilization (ENS; Est = −0.67, CI = −1.38–0.04, *p* = 0.065, Figure 2C) nor with richness of fungal species (observed genera; IRR = 0.36, CI = 0.11–1.17, *p* = 0.089, Figure 2D).

### 3.6. Detection and Growth of Oo Are Correlated with Microbial Community Composition

Independent extract replicates of each soil clustered strongly together in our NMDS plots, giving us confidence in our sequencing and data processing (Figure 3). Permutational analysis of variance demonstrated that the composition of fungal communities of experimental soils were significantly differentiated on the basis of whether the soil was collected from a hibernaculum or topsoil (PERMANOVA R^2^ = 0.29, F = 9.19, *p* = 0.001, Figure 3, Appendix A). Fungal community composition was also linked to the experimental growth of Oo (PERMANOVA, R^2^ = 0.09, F = 2.83, *p* = 0.01, Figure 3, Appendix A). There was no link between fungal community composition and environmental detection of Oo (PERMANOVA, R^2^ = 0.06, F = 1.97, *p* = 0.02, Figure 3, Appendix A).

Bacterial community composition differed between hibernaculum and topsoil samples (PERMANOVA, R^2^ = 0.47, F = 25.11, *p* = 0.001, Figure 3, Appendix A). Bacterial communities were not differentiated between soils in which Oo grew in experimental microcosms and those where it did not (PERMANOVA, R^2^ = 0.05, F = 2.47, *p* = 0.07, Figure 3, Appendix A). Bacterial community composition differed between samples in which Oo was detected and those in which it was not (PERMANOVA, R2 = 0.08, F = 4.73, *p* = 0.008, Figure 3, Appendix A).

Of the 301 unique bacterial and 379 unique fungal genera detected, hibernaculum soils and topsoils were differentiated by a total of 107 bacterial and 104 fungal biomarker genera (Appendix A, respectively). Soils in which Oo was environmentally detected were enriched with 19 bacterial genera and depauperate in 4 bacterial genera (Appendix A). Soils in which Oo grew in experimental microcosms were enriched with 13 fungal genera and depauperate in 14 fungal genera (Appendix A).

## 4. Discussion

Snake fungal disease is frequently observed in snakes emerging from hibernation, and it has been suggested that hibernation plays an important role in the ecology of the disease [15]. We found that Oo was significantly more likely to be detected in soils from within snake hibernacula compared to matched topsoil samples. Many North American snake species, including northern pine snakes, hibernate communally for several months in underground refugia; these snakes often return to the same hibernacula each winter [28,29,30,37]. High concentrations of infected snakes in a confined space for an extended period of time could result in the build-up of shed Oo in the surrounding soil, thus resulting in an increased likelihood of detection compared to soils sampled above ground where snakes are more mobile and dispersed.

Based on its physiological traits [19,38], it has been assumed that Oo is a saprotrophic fungus capable of proliferating in soils. However, experimental evidence demonstrating that growth occurs in soil substrates has hitherto been lacking. Our microcosm experiments confirmed that Oo can indeed grow in soil; however, this growth was conditional and perhaps not as ubiquitous as might be expected for a true saprotroph. Specifically, Oo grew in only a subset of sterile soil samples tested (*n* = 4 of 10). When the native microbial community remained intact, growth in soil was completely inhibited.

The microbial communities of soils have long been demonstrated to limit the ability of pathogenic organisms to form pathogen reservoirs [5,27,39,40]. Pathogen suppression in soils due to microbial community composition can be partitioned into general and specific pathogen suppression [39]. General suppression occurs because microbial communities as a whole sequester environmentally available resources, limiting the proliferation of an invading pathogen through competitive exclusion and denial of adequate resources [27]. Many pathogenic fungi are specialized to derive nutrients from their hosts [41]. This specialism may, in turn, reduce the ability of pathogenic fungi to compete with generalist species for resources in more complex environments with diverse microbial communities. The fact that Oo did not grow in any soil that possessed an active microbial community suggests that soils may demonstrate a high degree of general suppressive qualities toward Oo.

In contrast, specific pathogen suppression can be due to the presence of specific microbial species or taxa, which have an antagonistic relationship with the invading pathogen [39]. Such relationships can occur through the secretion of anti-microbial metabolic compounds that impact the pathogenic microbe, either directly or indirectly. We found that the composition of soil microbial communities was a significant predictor of Oo growth in paired sterilized soils. This presents the possibility that when barriers to infiltration, such as competition with native microbial communities, are removed, residual characteristics of those communities may still inhibit the growth of Oo in sterile soils. This could be due to the presence of stable secondary metabolites produced by microbes which remain biologically active after autoclaving [42]. Thus, our results indicate that soils may also demonstrate specific suppression and contain specific fungal taxa that produce metabolites which are inhibitory to Oo. In total, we detected 13 fungal genera which were significantly more abundant in soils that suppressed Oo growth. Even though the growth of Oo appears to be prevented via general suppression by whole microbial communities of soil, the viability of persistent Oo could be further impacted by specific suppression demonstrated by soils. The manipulation of soil microbial communities to increase the suppression of pathogens is commonplace in agriculture [27,43], and fungal species belonging to the differentially abundant genera that we observed could represent potential targets of future research aiming to develop bioaugmentation-based mitigation strategies of SFD.

With respect to detection of Oo in soils, our characterization of the microbial communities demonstrated that soils in which Oo was detected had increased microbial diversity compared to those in which it was not. The diversity of soil microbial communities has long been assumed to be positively correlated with the ability of a soil to inhibit survival of a pathogen [27]. However, microbial communities of soils are known to be influenced by a number of abiotic parameters including subsurface sampling depth [44]. We measured the pH, potassium, phosphorus and organic matter levels of the soils used in this experiment and found that hibernaculum soils possessed significantly lower levels of potassium and organic matter content than topsoil samples. Abiotic factors alone are known to affect persistence and suppress the growth of certain pathogenic microorganisms in soil [45]. However, neither the detection of Oo nor the ability of Oo to grow were statistically linked to any abiotic parameter that we measured in our soils, although this may have been due to small sample size. Given the correlational nature of the data that we collected, establishing any sort of cause-and-effect relationship between microbial diversity and the occurrence of Oo in our study is not possible because both factors may be independently linked to abiotic parameters of hibernaculum soils. Additionally, the soil parameter analyses that we conducted are those that the University of Wisconsin’s soil sciences extension consider fundamental to understanding the composition of a soil sample, but do not encompass the full range of possible soil elements, micronutrients, and physical characteristics. It is plausible that an unmeasured soil parameter may be linked to both soil microbial community composition and Oo growth/detection. Future studies should consider including a broader suite of soil parameter analysis.

It is important to note that growth and proliferation outside of a host are not required for the establishment of environmental reservoirs; instead, persistence of viable propagules in the soil may initiate infections [46]. The persistence of even low levels of a pathogen within an environment allows for cycles of disease that can drive population declines [47,48]. Despite an apparent inability for Oo to grow in the soils we tested, we did not detect declines in the amount of detectable DNA between time points. This could suggest that Oo conidia were persisting in the soils even if not actively growing. Low numbers of conidia plated for CFU counting means that we were only able to detect increases in viable Oo and not persistence of viable conidia over time. However, repeated and consistent environmental detections and the demonstrable ability to culture the fungus from the environment implicates hibernaculum soils as potential environmental reservoirs through the mechanism of pathogen persistence. Given that snakes congregate and spend prolonged periods of time within hibernacula while infected and demonstrate philopatry to hibernation sites [30], it is easy to see how hibernaculum soils may act as both recipient and donor of viable Oo conidia.

## 5. Conclusions and Future Directions

Our results, considered with the fact that all known isolates of Oo have been derived from snakes (or shed snakeskin; [49]), provide evidence that Oo is a specialized pathogen that is closely associated with snakes rather than a ubiquitous environmental saprotroph that acts as an opportunistic pathogen.

Specifically, we provide evidence that the environmental distribution of detectable Oo is associated with snake hibernacula. However, further research is needed to ascertain whether this distribution is driven by elevated persistence of Oo in hibernaculum soils or by the increased shedding of Oo by hibernating snakes. Further research will be required to fully evaluate the ability of Oo to persist in environmental sediments, irrespective of its proliferation and the environmental doses necessary to initiate infection.

We have also demonstrated that functional native microbial communities may inhibit the growth of Oo in soils. While this phenomenon could limit the infectious potential of environmental reservoirs of Oo, our sample size was small, and our test soils were collected from a relatively small geographical area that does not encompass the broad range of abiotic and biotic features of soils throughout the known range of Oo. This may also explain why we found little evidence that the growth or persistence of Oo is linked to abiotic soil parameters. Additionally, our samples were collected from snake hibernacula which are excavated annually. The mechanical disruption of soils has been shown to affect native microbial communities (e.g., [50]). As such, our results may not be indicative of snake hibernacula which remain undisturbed. Future experiments targeting a larger and more diverse sample set may help determine whether the differences we observed are meaningful for the occurrence and growth of Oo across its range. Nevertheless, our results suggest that it may be possible to mitigate SFD in wild snake populations through the manipulation of environmental microbial communities to enhance both general and specific suppression qualities. Such biological mitigation strategies would be a welcome development to North American snake conservation.

## Figures and Tables

**Figure 1 jof-07-00461-f001:**
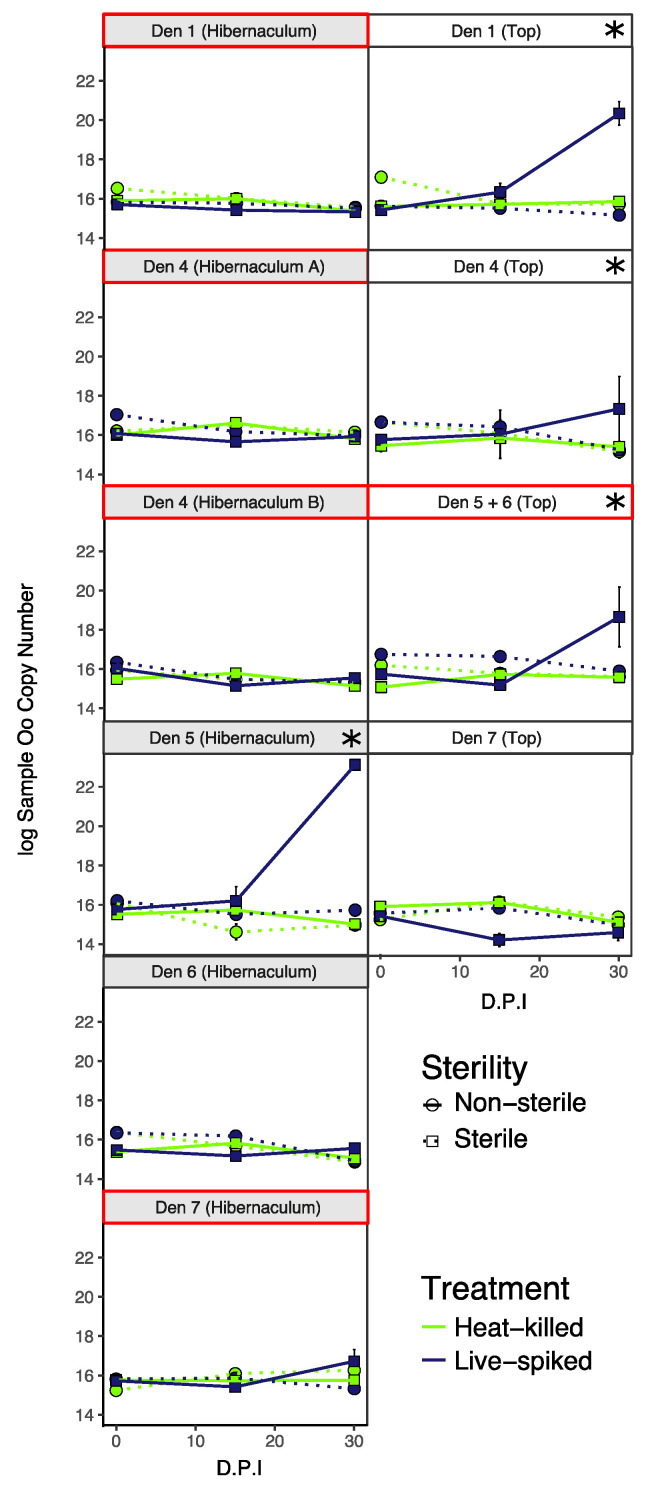
Growth curves of *Ophidiomyces ophidiicola* (Oo) in experimental microcosms. Plots in the right hand column represent topsoil samples; plots in the left hand column (grey header) represent hibernaculum soils. Red-header borders denote soils that were naturally qPCR positive for Oo. In order to aid interpretability of the plot, the Y-axis is the log of the qPCR target copy number of Oo present within each DNA extract. The X-axis is days post-inoculation (DPI). Microcosms spiked with heat-killed Oo are shown in chartreuse and those with live Oo in blue. Dashed lines connect those soils which were sterilized prior to inoculation and solid lines those which were not. Lines with an upward trajectory over time demonstrate an increase in Oo DNA (i.e., Oo growth). We determined Oo to have grown if there was greater than a 2 orders of magnitude difference between the mean Oo copy numbers of the heat-killed and live-spiked sample groups for a given soil and sterility treatment at a time point. Classified as such, growth of Oo was observable in four out of ten soils tested at 30 days post-inoculation (denoted by *). However, no growth was observed in any soils in which the native microbial community was left intact (non-sterile).

**Figure 2 jof-07-00461-f002:**
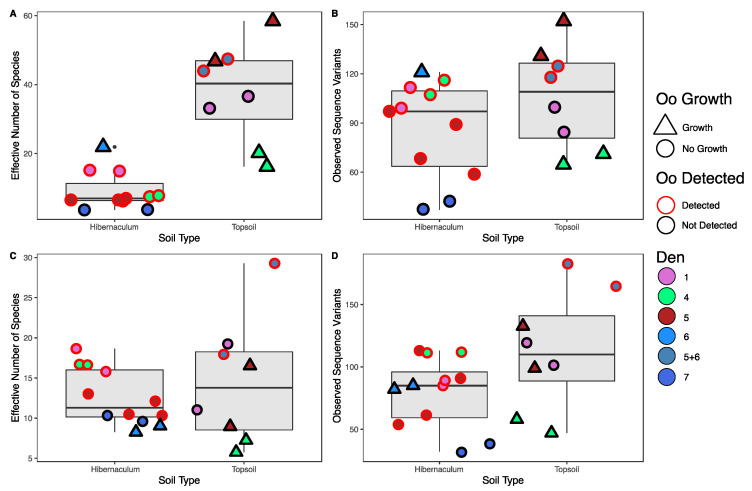
Alpha diversity plots of (**A**) bacterial communities of experimental soils based on effective number of species (ENS) scores; (**B**) bacterial communities of soils based on raw number of observed SVs; (**C**) fungal communities of experimental soils based on ENS scores; and (**D**) fungal communities of soils based on raw number of observed SVs. Each den is represented by a distinct color, and the left box plot of each plot displays soils collected from within hibernacula and the right displays associated topsoil samples. Soils in which Oo was detected prior to experimental inoculation are outlined in red, while those with no Oo detected are outlined in black. Soils in which Oo grew during experimental microcosm assays are represented by triangular points, while those in which it did not are represented by circular points. The diversity and richness of bacterial communities were lower in hibernaculum samples compared to topsoil samples. Bacterial diversity and richness were also significantly lower in soils which permitted Oo growth. Fungal diversity and richness were lower in hibernaculum samples than topsoil samples. Fungal diversity and richness were also higher in those soils in which environmental Oo was detected.

**Figure 3 jof-07-00461-f003:**
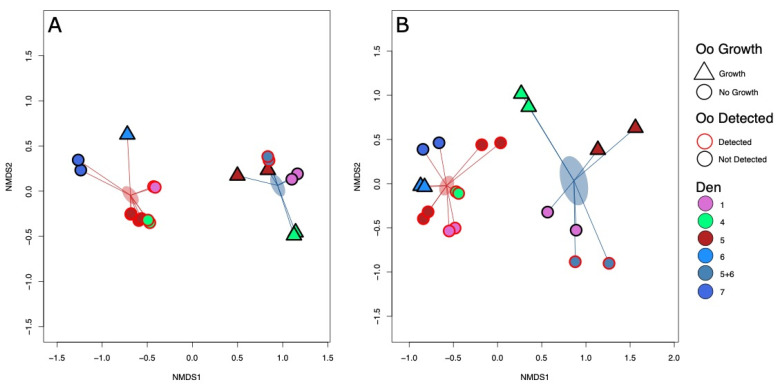
Non-metric multi-dimensional scaling ordinations based upon Bray–Curtis between sample dissimilarity metrics. (**A**) The bacterial communities of experimental soils; and (**B**) the fungal communities of experimental soils. Each den is represented by a distinct color. Soils that tested positive for Oo prior to experimental inoculation are outlined in red, while those that did not are outlined in black. Soils in which Oo grew during experimental microcosm assays are represented by triangular points, whilst those in which it did not are represented by circular points. Points are linked to the centroid of the respective soil type (hibernaculum vs. topsoil sample). Points linked by red lines to the red centroid represent hibernaculum soils. Those linked by blue lines to the blue centroid represent topsoil samples. Clear divergence can be seen between the composition of both the fungal and bacterial communities of topsoil samples and hibernaculum soils. Fungal communities also differed significantly based on Oo growth. Bacterial communities differed significantly based on Oo detection.

**Table 1 jof-07-00461-t001:** Detection of *Ophidiomyces ophidiicola* (Oo) in soil samples. The amount of Oo in each soil was assessed using qPCR. (A) Results of initial survey. Soils reported in bold text are those which produced viable Oo colonies when cultured. (B) Results of follow-up study which included sampling soil within snake dens/hibernaculum and adjacent topsoil samples. Den 4 is represented by two hibernaculum soil samples due to the presence of two distinct hibernation chambers and dens 5 and 6 are represented by a single topsoil sample due to their proximity. The amount of Oo in each sample was assessed using three replicate extractions, each quantified using a single qPCR. The cycle threshold (CT) values presented here represent the mean CT value for that soil sample. NA = no amplification of target amplicon.

**A—Initial Survey**			
**Soil name**	**Type**	**Oo qPCR CT Value**	**qPCR Interpretation**
Den 1 A	Hibernaculum	35.43	Detected
Den 1 B	Hibernaculum	37.42	Detected
Den 1 C	Hibernaculum	31.74	Detected
**Den 1 D**	**Hibernaculum**	**31.96**	**Detected**
Den 1 E	Hibernaculum	32.01	Detected
Den 1 F	Hibernaculum	35.78	Detected
**Den 1 G**	**Hibernaculum**	**31.6**	**Detected**
Den 1 H	Hibernaculum	NA	Not Detected
Den 2 A	Hibernaculum	35.68	Detected
Den 2 B	Hibernaculum	40	Detected
Den 3 A	Hibernaculum	34.87	Detected
Den 3 B	Hibernaculum	37.25	Detected
Den 3 C	Hibernaculum	NA	Not Detected
Den 3 D	Hibernaculum	35.55	Detected
Den 3 E	Hibernaculum	34.23	Detected
**B—Follow-Up Study**		**Mean CT**	
Den 1	Hibernaculum	34.86	Detected
Den 4 A	Hibernaculum	27.14	Detected
Den 4 B	Hibernaculum	32.66	Detected
Den 5	Hibernaculum	NA	Not Detected
Den 6	Hibernaculum	NA	Not Detected
Den 7	Hibernaculum	35.549	Detected
Den 1	Topsoil	NA	Not Detected
Den 4	Topsoil	NA	Not Detected
Den 5+6	Topsoil	39.19	Detected
Den 7	Topsoil	NA	Not Detected

## Data Availability

The raw data files associated with this study are available via a U.S. Geological Survey data release (https://doi.org/10.5066/P9MMWG10). Raw next-generation sequencing datasets associated with this work are available from the NCBI sequence read archive (Bioproject accession PRJNA691309). Associated analysis scripts are available from the GitHub repository github.com/zoolew/SnakeFungalDiseaseSoils.

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
