# Peer review of "Soil Reservoir Dynamics of Ophidiomyces ophidiicola, the Causative Agent of Snake Fungal Disease"

_jof, 2021, doi:10.3390/jof7060461_

Round 1

Reviewer 1 Report

Campbell et al. present an interesting study about Ophidiomyces ophiodiicola using soils as a reservoir. Overall, the introduction and discussion are well-written. However, the methods and results could be improved to help the reader understand the experimental design better. Below are a series of critiques by line. I have also attached a pdf document with highlighted comments and minor editorial suggestions. With some revisions this manuscript will be suitable for publication in Journal of Fungi. I applaud the authors for this interesting work on an understudied wildlife pathogen.

Line 107, 151 and 152 – uses the term den to describe hibernacula, but at line 157 says for the remainder of the manuscript den will be used for hibernacula and top soil. Would be easier for the reader to simply use the word hibernacula prior to line 157 to prevent confusion.

Line 110 Assemblage not community. Community implies interactions between the members.

Line 150 – Please consider creating a schematic diagram to show the sample collection and experimental design.

Lines 200 I think including a table would help visualize the combinations. It is hard to follow in the current state.

Lines 212 How is copy number determined? Relative to control or standard?

Line 213 I am unclear on what you mean by soil and sterility combination I think a table would help readers follow.

Line 220 what statistical analyses? Why include stats only in supplemental file? Is this a journal specific thing?

Line 224 Supplemental methods say v4 here it says v3.

Line 249 Pet peeve of mine, highly significant is just significant. P values are cut offs and the smaller they are is meaningless.

Line 257 – Mentions no evidence that detection or growth of Oo is related to soil parameters, but doesn’t show any analysis in manuscript or supplemental material to support this claim. There’s also no clear layout of what soil parameters were measured.

Figure 1 These color combinations will be difficult for color blind readers. I understand that the take home is the change in slope but why are all lines starting around 16 copies? Please see additional comments in the pdf document about this figure.  

Lines 284-287 Would there be an issue of autocorrelation with paired topsoil and hibernacula soil that would require something like a (G)LMM? This is likely an issue for all of the analyses. Probably will need to include this den ID as an effect in all analyses (PERMANOVA strata = denID)

Figure 2 Why are the dens shown? It makes the figure look overwhelming when trying to interpret the message conveyed by the plots. There is a lot going on in this and S2-3 which I think dilutes or distracts the readers from the take home message. I would suggest breaking this up into more graphs (or panels) with Oo growth and Oo detection being separate. Also denote significant differences with a letter or *.

Line 313 and Figure 3 Did you satisfy the assumptions of PERMANOVA, specifically the assumption of heteroskedasticity? It appears that within the nmds for fungal topsoil assemblages there is a lot more variation than for the hibernacula. Betadisper in package vegan can be used to test for this.

Line 322 PERMANOVA addresses differences in average assemblage composition rather than overall structure.

Line 440 Pleas tone the language down. In order for me to agree with this statement that Oo is strongly associated with snake hibernacula I would expect to see a larger sample size.

Line 446 Please tone the language down. You have found a correlation between microbial communities and Oo growth in soil. You can infer or hypothesize that interactions are causing this pattern but without actually addressing this in a manipulative study you cannot say for certain.

Minor

Line 34 why is Emerging in caps?

Lines 36-37 “and wild animals, and plant health”

Lines 80 and 90 italicize genus sp names

Lines 97 “of the where” doesn’t make sense grammatically

Line 130 remove the space in 4k Hz to 4 kHz

Line 142 change to  °C

Line 201 change 30m to 30 minutes since you use that elsewhere

Line 209 Bohuski et al is not in lit cited

Reviewer 2 Report

The authors investigate whether the fungus, Ophidiomyces ophidiicola (Oo), which is the causal agent of snake fungal disease, has an environmental reservoir in soils. To do this, they surveyed hibernaculum and adjacent topsoil for the presence of Oo using qPCR. In addition, they did Oo growth experiments in soil from different sources, using an experimental design where either alive or heat killed Oo were added to natural or autoclaved soil.  Lastly, they examined the bacterial and fungal communities in the soil samples in which Oo was grown. The authors found that Oo was often present in hibernaculum, and less common on adjacent topsoil.  Furthermore, Oo was unable to grow in soil samples that had natural microbial communities but did grow in samples that were autoclaved, which support the hypothesis that natural microbial communities in soil inhibit the growth of Oo. The authors propose using bioaugmentation of soil microbial communes as a mechanism to control Oo in hibernaculum.

Introduction:

From the first paragraph, it sounds like the plan is to test how Oo is transmitted and specifically whether Oo has an environmental reservoir. I think this can be made clearer in the last paragraph of the introduction by stating a more inclusive hypothesis that your other hypotheses fit under.

I would consider adding the soil paragraph after you introduce snake fungal disease. I see you are trying to make this a general paragraph about soil as a reservoir, but soil is only a reservoir for pathogens that have hosts that come in contact with soil.

Line 34: Is emerging suppose to be all in capital letters?

Line 39: is “surrounding” before environment necessary?  

Line 64: That rather than which

Line 90 and other spots. Shouldn’t the genus and species be italicized?

Methods:

Is there a better name than pilot study? Initial survey?

Line 114: More context would be beneficial to guide the reader. What are you testing with the pilot study? This seems to be done well in for the other method sections.

Line 132: What was used for the positive control/standard curve for the qPCR reactions?

Can the section on the pilot study be condensed?

Lines 223: It seems odd that none of the molecular methods for amplicon sequencing are referenced in the main text.

Line 257: Presumed formatting error

Table 1: Is there a reason to present a table with all of the samples rather than a summary of those results?

Figure 1: Red highlighted boxes indicate positive Oo via qPCR. For these experiments, do you know whether you are examining Oo that was already present in the soil vs what you used for the inoculations?  

Figure 1: What does it mean if you have a log copy number of the Oo ITS gene of between 14-16? Was there a treatment that had an inoculation that had no Oo at all (live or dead)?

Figure 2: From the table, it appears that none of the topsoil groups had Oo detected, this contradicts this figure where a topsoil sample had Oo detected. In the text it is clear that one top soils sample was positive.

Line 286: Is species diversity suppose to be ENS?

Line 368: extra space

Line 378: Why does this have to be specific? If bacteria A is antagonistic with bacteria B and interference competition occurs, then why can't the metabolites produced by the competition indirectly inhibit the pathogen?

Line 387: This is a bit unclear. Is it the soil that has specific suppression or the fungi?
